

PeerJ Hubs
Published on behalf of

International Association for Biological Oceanography
IABO

# Volcanic-associated ecosystems of the Mediterranean Sea: a systematic map and an interactive tool to support their conservation

Valentina Costa[1], Valentina Sciutteri[2], Pierpaolo Consoli[2], Elisabetta Manea[3], Elisabetta Menini[4], Franco Andaloro[5], Teresa Romeo[6,7] and Roberto Danovaro[8]

[1] Department of Integrative Marine Ecology (EMI), Stazione Zoologica Anton Dohrn, Amendolara, Italy
[2] Department of Integrative Marine Ecology (EMI), Stazione Zoologica Anton Dohrn, Messina, Italy
[3] Laboratoire d'Ecogéochimie des Environnements Benthiques (LECOB), Sorbonne Université, Banyuls sur Mer, France
[4] Nicholas School of Environment, Duke University, Beaufort, NC, United States
[5] Department of Integrative Marine Ecology (EMI), Stazione Zoologica Anton Dohrn, Palermo, Italy
[6] Department of Integrative Marine Ecology (EMI), Stazione Zoologica Anton Dohrn, Milazzo, Italy
[7] National Institute for Environmental Protection and Research, Milazzo, Italy
[8] Department of Life and Environmental Sciences, Polytechnic University of Marche, Ancona, Italy

Corresponding author
Valentina Costa,
valentina.costa@szn.it

## ABSTRACT

**Background:** Hydrothermal vents, cold seeps, pockmarks and seamounts are widely distributed on the ocean floor. Over the last fifty years, the knowledge about these volcanic-associated marine ecosystems has notably increased, yet available information is still limited, scattered, and unsuitable to support decision-making processes for the conservation and management of the marine environment.
**Methods:** Here we searched the Scopus database and the platform Web of Science to collect the scientific information available for these ecosystems in the Mediterranean Sea. The collected literature and the bio-geographic and population variables extracted are provided into a systematic map as an online tool that includes an updated database searchable through a user-friendly *R-shiny* app.
**Results:** The 433 literature items with almost one thousand observations provided evidence of more than 100 different volcanic-associated marine ecosystem sites, mostly distributed in the shallow waters of the Mediterranean Sea. Less than 30% of these sites are currently included in protected or regulated areas. The updated database available in the *R-shiny* app is a tool that could guide the implementation of more effective protection measures for volcanic-associated marine ecosystems in the Mediterranean Sea within existing management instruments under the EU Habitats Directive. Moreover, the information provided in this study could aid policymakers in defining the priorities for the future protection measures needed to achieve the targets of the UN Agenda 2030.

## INTRODUCTION

Hydrothermal vents, cold seeps, pockmarks, seamounts and mud volcanoes are typically found within volcanic areas. By definition, hydrothermal vents are formed when seawater percolates through fissures in the ocean crust near spreading centers or subduction zones, becomes heated by hot magma and then emerges to form vents (*Rizzo et al., 2022*). Cold seeps are areas where hydrogen sulfide, methane and other hydrocarbon-rich fluid seep out of the ground, often in the form of a brine pool. The temperature of these seeps is often slightly higher than the surrounding water (*Vanreusel et al., 2009*). Seamounts are underwater mountains generally formed by volcanic activity that in the case of mud volcanoes can spew mud, gas and fluid (*Dimitrov, 2002*), while pockmarks are deep depressions in sediments caused by the escape of gas from beneath the seafloor (*Cathles, Su & Chen, 2010*).

In 1977, the discovery of the first hydrothermal vents in the Galapagos Rift (*Corliss et al., 1979*) has had a significant impact on theories about the origins of life, with microorganisms in the depths using hydrogen sulfide to live and grow (*i.e.*, chemosynthesis) instead of using lights to synthetise organic material (*i.e.*, photosynthesis) (*Van Dover, 2000*). Soon after the discovery of hydrothermal vents in the Galapagos Rift, similar volcanic-associated ecosystems have been found in every ocean basin with extremely variable environmental conditions, from shallow to hadal (>6,000 m) depths, even in the Antarctic regions (*Domack et al., 2005*). During the last decades the more intense observation of the European continental margins provided evidence of the presence of a wide range of volcanic-associated ecosystems such as hydrothermal vents, cold seeps, pockmarks, seamounts and mud volcanoes in the Mediterranean basin (*Loncke, Mascle & Fanil Scientific Parties, 2004*). Overall, the biological processes and the diverse communities of volcanic-associated ecosystems are unique and our understanding remains limited (*Van Dover, 2000*; *Vanreusel et al., 2009*; *Aiello et al., 2022*).

There are numerous international initiatives that highlight how the importance of these ecosystems grows with the advancement of their discovery. The "InterRidge Global Database of Active Submarine Hydrothermal Vents Fields" (Version 3.4, *Beaulieu & Szafrański, 2020*) lists over 550 active submarine hydrothermal vent fields worldwide, including 21 in the Mediterranean Sea (*Beaulieu & Szafrański, 2020*). Thanks to the diffusion of Side Scan Sonar, Remotely Operated Vehicles (ROVs), and the advances in sampling equipment, additional sites have been discovered (*Martorelli et al., 2016*; *Donnarumma et al., 2019*; *Saroni et al., 2020*). The "InterRidge Workshop on Management and Conservation of Hydrothermal Vent Ecosystems" has developed the *Criteria for Identifying Critical or Sensitive Sites* based on their scientific value or significance for species survival, as well as a *Code of Conduct* to minimize conflicts and environmental impacts (*Dando & Juniper, 2001*). The *Code of Conduct* has been adopted by the OSPAR

Convention (*OSPAR Commission, 2008*), and several authors have continued to advocate for the protection of these chemosynthetic environments (*Taviani, 2014*; *Van Dover et al., 2018*; *Esposito et al., 2018*; *Consoli et al., 2021*).

Internationally, hydrothermal vents and seamounts are included among the "Threatened" and/or "Declining Species and Habitats" (Oslo-Paris Convention for the Protection of the Marine Environments of the North-East Atlantic, List of Threatened and/or Declining Species and Habitats, Agreement 2008-06, *OSPAR Commission, 2008*). They are considered "*reef*" to be preserved according to the EU Habitats Directive (92/43/EEC, *European Union, 1992*), and are listed as Vulnerable Marine Ecosystems (VMEs) by the Food and Agriculture Organization of the United Nations (FAO) and the Regional Fisheries Management Organizations based on their "*vulnerability*" and fragility against damage from bottom trawling (*United Nations General Assembly, 2006*; *Food and Agriculture Organization of the United Nations (FAO), 2009*, *2019*).

In the Mediterranean Sea, several shallow hydrothermal vents systems such as those in Castello Aragonese (Ischia Island, Italy), Levante Bay (Vulcano Island, Italy), Panarea Island (Italy) and Palaechori Bay (Milos Island, Greece) (*e.g.*, *Thiermann et al., 1997*; *Caramanna, Espa & Bouché, 2010*; *Boatta et al., 2013*; *Rizzo et al., 2022*) are characterized by elevated $pCO_2$ and low pH values and are used as models for studying the effects of ocean acidification on marine organisms and ecosystems (*Aiuppa et al., 2021*). Cold seeps and pockmarks, which release gas and fluids often oxidized to carbon dioxide, also may offer insight into changes in the ocean chemistry including ocean acidification (*Judd et al., 2002*; *Olu-Le Roy et al., 2004*; *Joseph, 2017*). Seamounts and mud volcanoes can also provide valuable information about the ocean's circulation and climate (*Olu-Le Roy et al., 2004*). They can affect ocean currents and water masses, influencing the ocean's ability to store and transport heat and carbon. Despite the recognized importance of volcanic-associated ecosystems as key habitats, as documented by international initiatives such as the Convention on Biological Diversity through the designation of some deep regions as Ecologically or Biologically Significant Marine Areas (*Fanelli et al., 2021*), just a few European initiatives currently in place for protecting the Mediterranean marine environment include these ecosystems. For instance, the European Habitat Directive (92/43/ECC) list hydrothermal vents and pockmarks among the habitat to be protected as they harbor unique and diverse communities, provide important ecological services and have significant cultural, economic and scientific importance (*Tarasov et al., 2005*; *Price & Giovannelli, 2017*; *Caccamo et al., 2018*; *Lu et al., 2020*). Therefore, to fully understand and protect the ecological, scientific and economic relevance of marine hydrothermal vents, cold seeps, pockmarks, and seamounts in the shallow and deep Mediterranean Sea, comprehensive protection measures and management plans are necessary, along with extensive scientific research.

Although several studies have been conducted, our knowledge is still incomplete as information on these ecosystems are scattered and there is a general lack of awareness of their spatial distribution, which hamper potential *ad hoc* conservation planning. The production of evidence-based maps or Systematic Maps (*i.e.*, based on evidence from the literature, *sensu James, Randall & Haddaway, 2016*; *McKinnon et al., 2016*) is

increasingly needed to fill these gaps and provide spatially-explicit knowledge frameworks to feed environmental management and conservation purposes (*Randall & James, 2012*; *Haddaway et al., 2016*).

This study answers the following questions:

i) What evidence exists on volcanic-associated ecosystems including hydrothermal vents, cold seeps, pockmarks, seamounts and mud volcanoes in the Mediterranean Sea?

ii) How many volcanic-associated ecosystems are present in the Mediterranean Sea and where are they?

To address these questions, we developed a systematic map (SM) to (1) identify the available scientific literature related to these ecosystems in the Mediterranean Sea, (2) categorize and compare the scientific information available from different Mediterranean regions in the form of variables extracted from the literature (*e.g.*, geographic information and population analysed in the literature), (3) create a user-friendly and interactive map connected to a searchable database to support the translation of science into policy and management actions. By adopting a SM approach, this study gathers, categorizes and summarizes the available knowledge on volcanic-associated ecosystems in the Mediterranean Sea. Overall, we aimed to offer a tool to guide future research efforts and conservation initiatives in the Mediterranean Sea.

## MATERIALS AND METHODS

### Search strategy

The SM was created following the guidelines proposed by the CEE (*Collaboration for Environmental Evidence, 2022*) and the Reporting standards for Systematic Evidence Syntheses (ROSES) (*Haddaway et al., 2018*). CEE identifies a series of steps to follow in order to provide the quality standard and increased transparency and to allow reproducibility of the entire process (*Collaboration for Environmental Evidence, 2022*). ROSES was specifically designed for environmental management and conservation studies as a checklist that addresses all relevant methodological information that should be reported in the SM (*Haddaway et al., 2018*) (Table S1).

### *Scoping and keyword string definition*

Different search strings were tested to identify the most appropriate database for the literature data analysis. We started with a scoping stage based on the keywords "hydrothermal vents" AND "Mediterranean sea". To help define the keywords, we used an adaptation of the Population, Intervention, Control, Outcome (PICO) framework (*Collaboration for Environmental Evidence, 2022*). Here we used PICo (Population, Interest, Context), identifying the volcanic-associated ecosystems as our population, the evidence existing in the literature as our Interest and the Mediterranean Sea as the Context.

The final literature search included three substrings connected using Boolean operators (AND and OR) and the wildcard "*" (Table S2):

'("*hydrothermal**" OR "*emission**" OR "*volcan**" OR "*plume**" OR "*vent**" OR "*seep**" OR "*eruption**" OR "*acidification*" OR "*carbon dioxide*" OR "*pH*" OR "*CO$_2$*" OR "*CCS*") AND ("*Mediterranean*") AND ("*sea*" OR "*ocean*" OR "*marine*")'.

The substrings used were broad enough to collect a large amount of literature however limiting the results in line with the objective of the research (*James, Randall & Haddaway, 2016*). The final search string was slightly modified depending on the database used (Table S2). Finally, we specifically searched for the scientific articles directly related to some of the projects focused on volcanic-associated ecosystems carried out totally or partially in the Mediterranean Sea: MedSeA (Mediterranean Sea Acidification in a changing climate, 2011–2014, http://medsea-project.eu/), EPOCA (European Project on Ocean Acidification, 2008–2012, https://cordis.europa.eu/project/id/211384), BIOACID I, BIOACID II and BIOACID III (Biological Impacts of Ocean Acidification, 2009–2012, 2012–2015, 2015–2017, https://www.bioacid.de), MIDAS (Managing Impacts of Deep-seA reSource exploitation, 2013–2016, https://www.eu-midas.net/), HERMES (Hotspot ecosystem research on the margins of European seas, 2005–2009, https://cordis.europa.eu/project/id/511234) and HERMIONE (Hotspot Ecosystem Research and Man's Impact on European seas, 2009–2012, https://cordis.europa.eu/project/id/226354) (Table S2).

### Database and searches

The Scopus database and the platform Web of Science were used to collect the scientific literature. These databases were chosen to ensure the reliability of the gathered information, since only indexed and peer-reviewed publications are allowed, disregarding all the non-indexed works. It should be noted that only English language literature was searched for and retained. Reviews were retained only when new results were presented, and the reported references were analyzed to include missing studies in the SM.

An additional search was performed on Google Scholar, and with a screening of the first 100 results (*Haddaway et al., 2015*) (Table S2).

### Exported results

Search results from Scopus and Web of Science were exported in *.csv* format along with all the literature information including abstract and keywords. Literature search results from Google Scholar were manually added to a spreadsheet and all exported files were then loaded in the R environment (version 4.2.2; *R Core Team, 2022*) in RStudio (version 2022.12.0; *RStudio Team, 2022*) using the *revtools* package (*Westgate, 2019*).

### Duplicate removal

As literature searches were performed on multiple online tools, some publications (from now on referred to as items) might be present multiple times. Therefore, the DOI (Digital Object Identifier) was used to identify and remove duplicates using the function *find_duplicates* in the *revtools* package and to create a database of unique studies.

## Article screening and inclusion criteria

To produce the SM, the database was then screened following a set of selection criteria:

1. studies performed in the Mediterranean Sea;

**Table 1 Coding strategy used for extracting data from each study.**

| Category variables | Data extracted |
|---|---|
| Literature | Author(s), year, title, journal, DOI, abstract, author's keywords, index keywords, publication type |
| (Bio)geographic | Latitude, longitude, site, country, area, depth, site type |
| Population(s) | Environmental or biological target, target category, target group, target species |
| Response(s) | Survival, calcification, growth, photosynthesis and reproduction |

**Note:**
Data were extracted from the full-text reading and categorized based on four categories. The data extracted are reported for each category.

2. studies including the following habitat categories: hydrothermal vents, cold seeps, pockmarks, gas emissions areas;

3. studies based on *in situ* experiments or sampling studies analyzing environmental characteristics and/or biological aspects.

The first screening was based on title and abstract reading, performed using the function *screen_abstracts* in the *revtools* package. The next step involved the full-text retrieval using the different access provided by the co-authors and the creation of a library database of all studies using the open-source software Zotero (https://www.zotero.org).

## Consistency checking

A random selection of 100 items from the literature searches was carried out and screened by two authors. The *kappa statistic* was calculated to quantify the consistency between authors (*Collaboration for Environmental Evidence, 2022*), obtaining a value of 0.78. According to the classification by *Viera & Garrett (2005)*, a kappa ranging between 0.61 and 0.80 indicates "substantial agreement" between the two authors.

## Data coding and analysis

Literature, bio-geographic and population variables as well as biological responses were extracted from the full-texts reading. For studies that involved biological organisms, we categorized the type of response variable that was analyzed in the study, using the most common responses measured in literature such as: calcification (or dissolution), reproduction, growth, photosynthesis and survival (or mortality) (Table 1).

A single observation was defined as each single biological or environmental target within a literature item. Each observation is identified by a single group, species and response within a literature study and is recorded in a separate row in a *.csv* spreadsheet, with each variable given its column. Frequencies are then analyzed. Moreover, we extracted the keywords identified by the authors for each selected article and we performed a keyword frequency and co-occurrence analysis. The network of co-occurrences was analyzed and visualized. The textual analysis on keywords has proven to be an efficient and effective way to identify patterns, trends, gaps and relationships in large sets of unstructured data across scientific disciplines, such as marine ecology (*Fanini et al., 2021*).

The analyses were performed using *tidytext* (*Silge & Robinson, 2016*) and *widyr* packages (*Robinson, 2021*).

With the help of the *Shiny* libraries, the entire database was made freely accessible in the form of a *Shiny*-based application. *Shiny* is a R package that allows the creation of interactive maps in the form of an app, by combining the advantages of the computational power of R with the attractiveness and the easy handling of the web system (*Chang et al., 2021*). *Shiny* apps were originally designed for small applications consisting of two main entities: the *Shiny User Interface (SUI)* which provides all the aesthetic components the user interacts with, and *the Shiny Server Side (SSS)* which performs the required computations. The user interface of the shiny-based application has been implemented using the *shiny* package, and the graphical part of the application has been implemented through the functionalities of the *tidyverse* packages (*Wickham et al., 2019*).

## Data quality and confidence

The study type was specified in the data extracted from the full-text reading, giving some indication of the assessment of quality (*James, Randall & Haddaway, 2016*; *Collaboration for Environmental Evidence, 2022*). However, we did not explicitly assess the quality of each article as this step is considered optional in systematic mapping (*James, Randall & Haddaway, 2016*).

# RESULTS

## Systematic map results

### Search results and screening

Overall, 10,310 items were identified from the online database searches: 5,472 and 4,838 from the Scopus and Web of Science searches, respectively. In addition, the searches of scientific articles related to projects (MesSeA, EPOCA, BIOACID I, BIOACID II and BIOACID III, MIDAS, HERMES and HERMIONE) returned 858 items. Across the combined searches, a duplication rate of 36.4% was estimated and the number of remaining articles was 7,100. The title and abstract screenings excluded 5,691 articles. All the remaining literature items were retrieved (using the access provided by all the co-authors' institutions) and the full-text screening was then carried out. Finally, 433 literature items were coded in the SM database with a total of 992 observations (last update 27/05/2022; Fig. S1). The complete list of literature items included in the database is provided in the Supplemental Material (Table S3).

### Dates, study types and journals analysis

The earliest article in the database refers to a work published in 1973, followed by an increasing trend in the number of publications showing a peak in 2014 (Fig. 1).
The selected studies included four document types, with a total of 419 articles (96.8% of total studies), three conference papers (0.7%), eight proceedings papers (1.9%) and three reviews (0.7%) (Fig. 1). The studies were published in more than 100 different journals with only five journals that published more than 20% of the total: Marine Geology (8.6%), Chemical Geology (4.4%), Marine Environmental Research (3.0%), Science of the Total

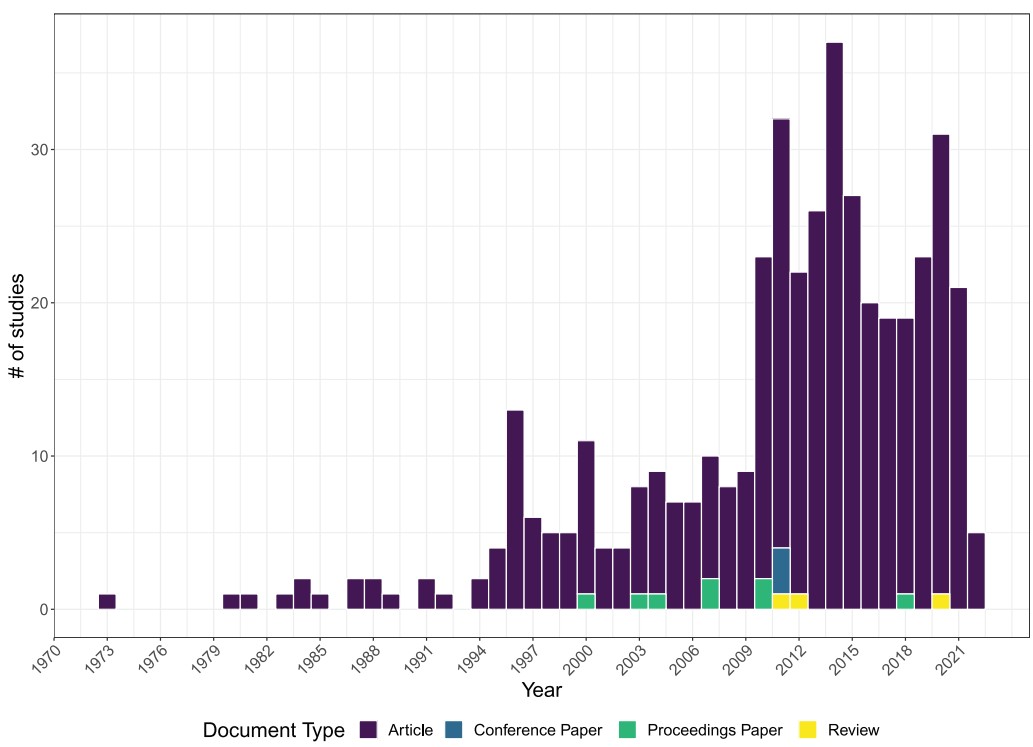

**Figure 1 Temporal trend of the literature items included in the database ($n$ = 433).** The literature items are categorized based on the type of document from 1973 to 2022.

Environment (3.0%) and Deep-Sea Research Part I: Oceanographic Research Papers (2.8%) (Fig. 2).

### (Bio)geography and population analysis

There are twenty-one regions in the Mediterranean Sea that include hydrothermal vents, cold seeps, mud volcanoes, pockmarks and seamounts with 156 unique sites between 1 and 3,800 m depth (Fig. 3). A list of all the sites is reported in Table S4.

Almost 50% of the total number of observations was located in the Italian Maritime Region ($n$ = 486), followed by Greece ($n$ = 221) and Turkey ($n$ = 121). Less than 30% of the observations ($n$ = 279) fall within some kind of protection measures by European, National or Regional regulation. The highest number of scientific observations was registered in the "Aeolian Archipelago, terrestrial and marine areas", a region identified as a Special Protection Area (SPA) under the EU Birds Directive (2009/147/EC, *European Union, 2009*) ($n$ = 166) (Fig. 3 and Table 2).

The highest fraction of observations identified falls into Hydrothermal Vents (47.9%), followed by Mud Volcanoes (30.7%) and Cold Seeps (5.8%) (Fig. 4A; Table S5). A slightly higher number of observations was reported in shallow water (<200 m depth) than in deep sea (>200 m depth; Fig. 4B).

Three regions included more than 50% of the total number of observations with the highest number recorded in the Aeolian Arc (27.9%), followed by the Gulf of Naples (14.7%) and the Aegean Arc (11.7%) (Fig. 4C).

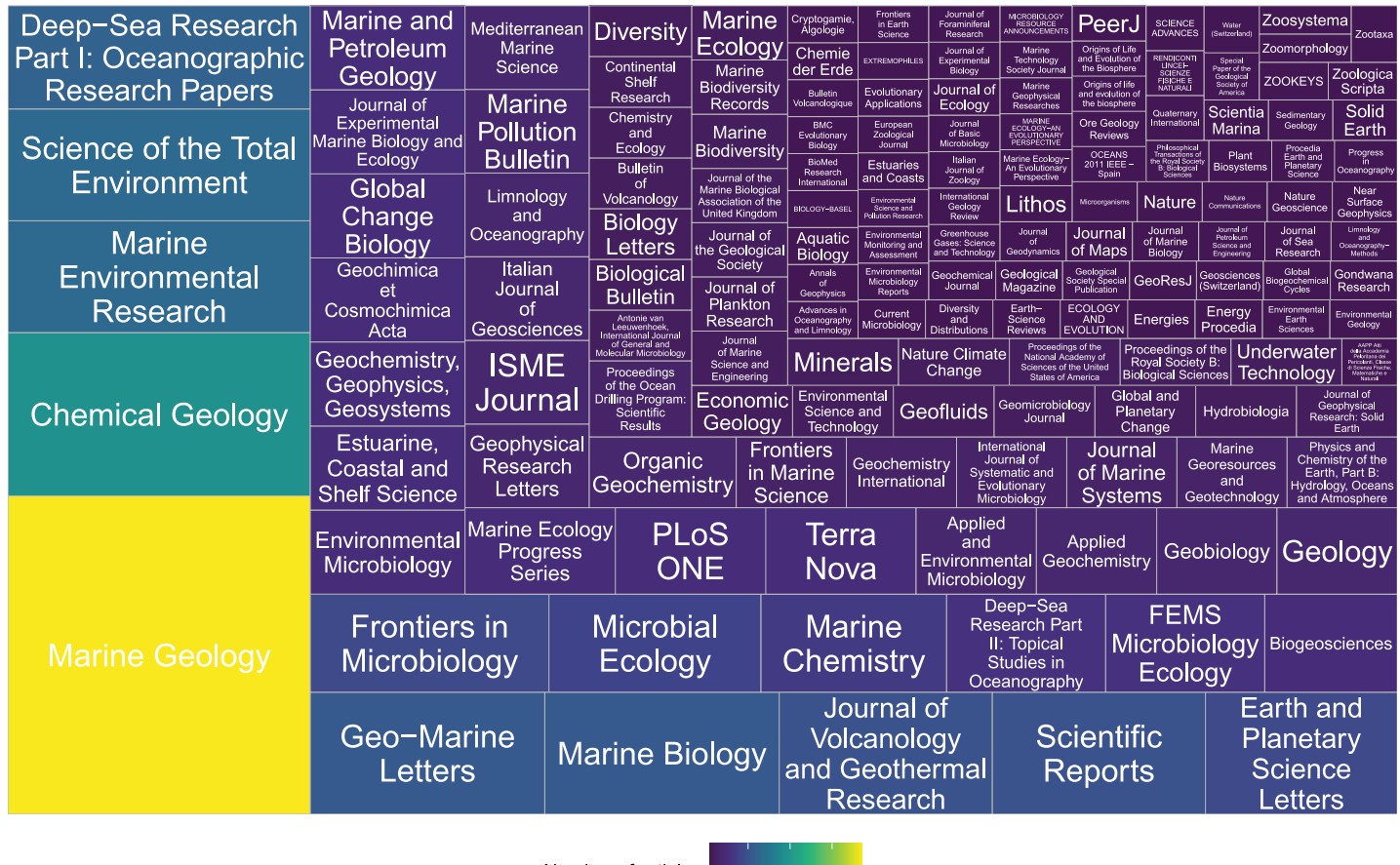

**Figure 2 Number of articles published by scientific journals (n = 433).** The hierarchical visualization of the number of studies was realized using the *treemapify* package (*Wilkins, 2021*). In the treemap, the size and colors of each tile are proportional to the number of published studies.

Generally, more attention was given to the environmental characteristics of the habitat (57.8%) than the biological aspects (42.2%), but a higher number of biological observations was recorded in shallow water than in deep sea (n = 284 and n = 133, respectively, Fig. 4D).

The highest number of environmental observations involved the analysis of sediment (56.9%), followed by water (14.0%) and gas (2.3%), with the rest of observations involving a combination of the three abiotic matrices (Fig. 4E). In the Aeolian Arc a higher number of observations involved the analysis of environmental characteristics (54.3%) than the biological components of the systems (45.6%).

Ten main biological targets were identified in the literature analyzed: algae, bacteria, epibenthos, epiphytes, fish, macrobenthos, meiofauna, plankton, seagrass and virus. The greatest number of observations was focused on macrobenthos (39.1%), followed by bacteria (26.6%) and algae (10.3%). The highest number of observations within the analyzed categories was related to shallow-water areas, with more than twice the number of observations for deep-sea areas (Fig. 4F). The highest number of biological observations involved the survival response (83%), with the other responses covering less than 20% of the total observations (Fig. 4G).

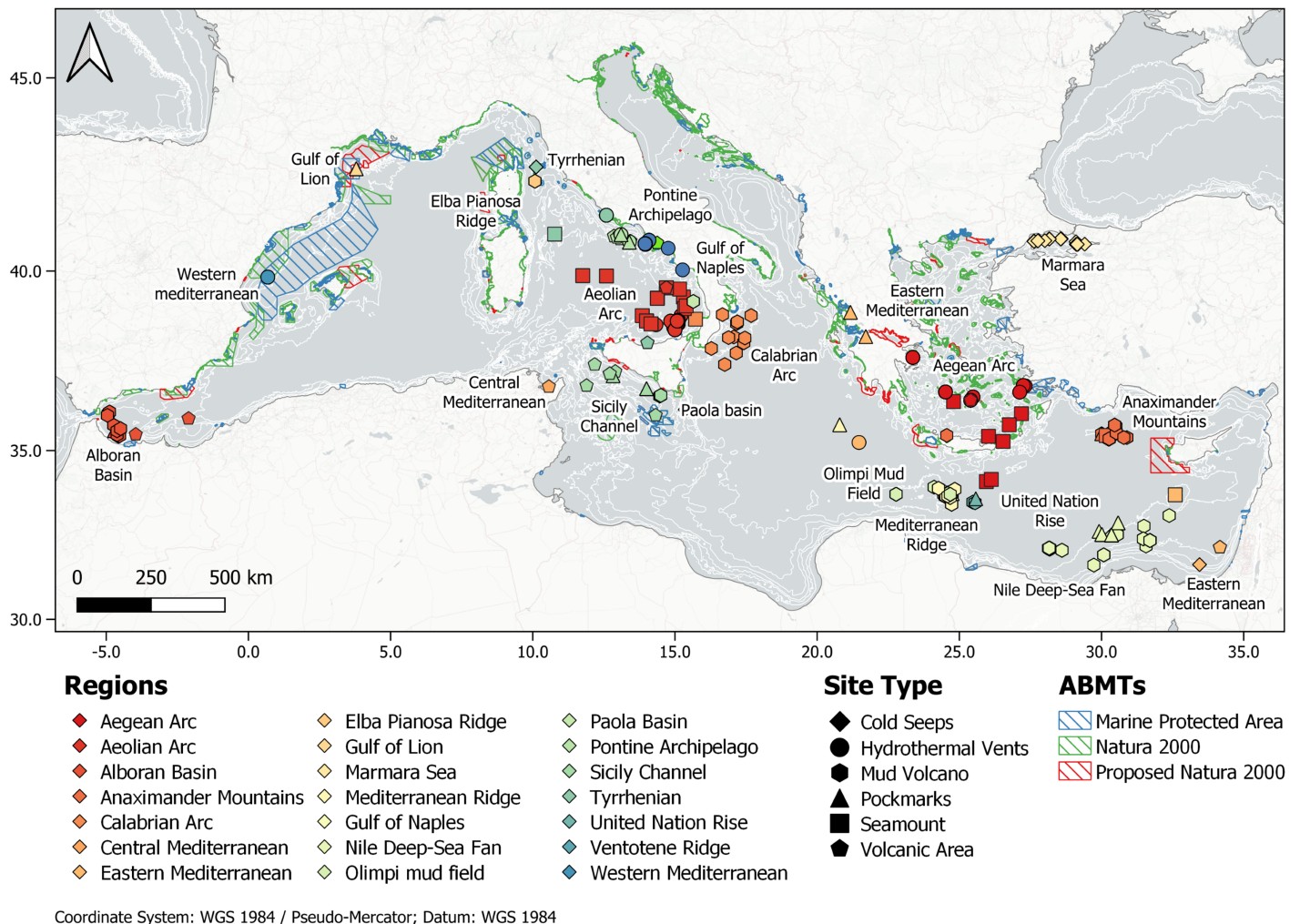

Coordinate System: WGS 1984 / Pseudo-Mercator; Datum: WGS 1984
Service Layer Credits: MAPAMED 2019 edition © 2020 by SPA/RAC and MedPAN; ETOPO Global Relief Bathymetry Data.

**Figure 3 Location of the scientific observations included in the database ($n$ = 992).** Polygon data of marine protected areas (MPA), Natura 2000 site and proposed Natura 2000 site (blue, green and red polygons, respectively) were modified from MAPAMED, the database of marine protected areas in the MEDiterranean. 2019 edition. © 2020 by SPA/RAC and MedPAN. Licensed under CC BY-NC-SA 4.0. Available at: https://www.mapamed.org/. Bathymetry data were obtained from the 1-min gridded global relief data ETOPO1 (2009, https://www.ngdc.noaa.gov/mgg/global/global.html). The map was generated using QGIS 3.24.1.

*Keywords analysis*

More than 900 unique keywords in the database were identified. The most frequent keywords were "ocean acidification" ($n$ = 53), "mud volcano" ($n$ = 35) and "mediterranean sea" ($n$ = 34), followed by "cold seep" ($n$ = 29) and "hydrothermal vent" ($n$ = 16), while all the other keywords were less mentioned in the whole database (Fig. 5A).

We also examined the results of the co-occurrences of author's keywords using a network visualization graph to identify patterns of relationships between keywords. The keywords "mediterranean sea" and "ocean acidification" mostly frequently occur together creating a cluster at the centre of the representation. In contrast, keywords related to "authigenic carbonates" in the "Sea of Marmara", "metalliferous sediments" in the "Aeolian Arc" or "microbial biofilms" remained at the margin of the network (Fig. 5B).

**Table 2 Denomination of AMBTs and number of observations that fall within.**

| Denomination | Natura 2000 network | Name | Country | # of obs | Site type | Maximum depth (in m) |
|---|---|---|---|---|---|---|
| International significance natural marine area | | Santuario Per I Mammiferi Marini | ITA | 2 | Mud volcano | 12 |
| | | | | 3 | Cold seeps | 12 |
| Marine nature park | | Golfe Du Lion | FRA | 2 | Pockmarks | 288 |
| Marine protected area | | Corredor de Migracion de Cetaceos del Mediterraneo | ESP | 3 | Hydrothermal vents | 40 |
| | | Isole di Ventotene e Santo Stefano | ITA | 1 | Hydrothermal vents | 80 |
| | | | | 2 | Volcanic area | 100 |
| | | Regno di Nettuno | ITA | 6 | Hydrothermal vents | 5 |
| Marine reserve | | Illes Columbretes | ESP | 3 | Hydrothermal vents | 40 |
| National park | | Parco Nazionale Del Cilento E Vallo Di Diano | ITA | 1 | Hydrothermal vents | 630 |
| National park—peripheral zone | | Periochi Perivallontikou Elegchou Ethnikou Parkou Ygrotopon Amvrakikou (Zoni C) | GRC | 1 | Pockmarks | 36 |
| Proposed site of community importance (Habitats directive) | | Récifs des canyons Lacaze-Duthiers, Pruvot et Bourcart | FRA | 2 | Pockmarks | 288 |
| | | THALASSIA PERIOCHI KOLOUMVO | GRC | 6 | Hydrothermal vents | 495 |
| | | | | 2 | Volcanic area | 495 |
| Regional/provincial nature reserve | | Riserva Naturale Orientata/Integrale Isola Di Stromboli E Strombolicchio | ITA | 4 | Hydrothermal vents | 100 |
| Sites of community importance (Habitats Directive) | | Żona Fil-Baħar Bejn Il-Ponta Ta' San Dimitri (Għawdex) U Il-Qaliet | MLT | 1 | Volcanic area | 180 |
| SPA (Birds directive) + pSCI (Habitats directive) | ESZZ16010 | Espacio marino del entorno de Illes Columbretes | ESP | 3 | Hydrothermal vents | 40 |
| SPA (Birds directive) + pSCI (Habitats directive) | IT8030010 | Fondali marini di Ischia, Procida e Vivara | ITA | 26 | Hydrothermal vents | 5 |
| Special area of conservation—international importance | MT000015 | Marine area between San Dimitri Point (Gozo) and Il-Qaliet | MLT | 1 | Volcanic area | 180 |
| Special area of conservation (Habitats directive) | GR2110001 | AMVRAKIKOS KOLPOS, DELTA LOUROU KAI ARACHTHOU (PETRA, MYTIKAS, EVRYTERI PERIOCHI, KATO POUS ARACHTHOU, KAMPI FILIPPIADAS) | GRC | 1 | Pockmarks | 36 |
| | IT6040020 | Isole di Palmarola e Zannone | ITA | 3 | Volcanic area | 150 |
| | IT8050008 | Capo Palinuro | ITA | 1 | | |
| | IT6000016 | Fondali circostanti l'Isola di Ponza | ITA | 2 | Volcanic area | 105 |
| | IT6000018 | Fondali circostanti l'Isola di Ventotene | ITA | 2 | Volcanic area | 100 |
| | | Fondali Marini di Baia | ITA | 12 | Hydrothermal vents | 15 |
| | GR4210008 | KOS: AKROTIRIO LOUROS—LIMNI PSALIDI—OROS DIKAIOS—ALYKI—PARAKTIA THALASSIA ZONI | GRC | 2 | Hydrothermal vents | 4 |
| | | | | 1 | Volcanic area | 4 |
| | GR4210007 | NOTIA NISYROS KAI STRONGYLI, IFAISTIAKO PEDIO KAI PARAKTIA THALASSIA ZONI | GRC | 2 | Hydrothermal vents | 2 |

(Continued)

| Table 2 (continued) | | | | | | |
|---|---|---|---|---|---|---|
| Denomination | Natura 2000 network | Name | Country | # of obs | Site type | Maximum depth (in m) |
| Special protection area (Birds directive) | | Żona Fil-Baħar Madwar Għawdex | MLT | 1 | | |
| | ITA030044 | Arcipelago delle Eolie—area marina e terrestre | ITA | 166 | Hydrothermal vents | 1,100 |
| | IT8050008 | Capo Palinuro | ITA | 1 | | |
| | IT6040019 | Isole di Ponza, Palmarola, Zannone, Ventotene e S. Stefano | ITA | 1 | Hydrothermal vents | 80 |
| | | | | 7 | Volcanic area | 150 |
| | MT0000112 | Marine area around Gozo | MLT | 1 | Volcanic area | 180 |
| Specially protected areas of Mediterranean importance SPAMI (Barcelona convention) | | Illes Columbretes | ESP | 3 | Hydrothermal vents | 40 |
| | | Pelagos sanctuary for the conservation of marine mammals | FRA; ITA; MCO | 2 | Mud volcano | 12 |
| UNESCO-MAB biosphere reserve | | Cilento and Val de Diano | ITA | 1 | Hydrothermal vents | 630 |

**Note:**
The denomination of the AMBTs, the Nature 2000 network ID, name and country are indicated, with the number of observations falling within the type of site identified and the maximum depth (in meters).

### A tool for managers: the MH-shiny app and its interactive map

In the context of the FAIR principles (Findable, Accessible, Interoperable and Re-usable data) of the European Commission "Open Data Directive" (*European Union, 2019*), the *MH-shiny* has been developed during this study (*cfr. link*: https://costavale.shinyapps.io/MH-shiny/) as a *shiny*-based application freely accessible online. The complete R code and the data are freely available on a GitHub repository (https://github.com/costavale/MH-shiny/; DOI 10.5281/zenodo.7537047). The *shiny* app works on both local and online versions on macOS, Windows, and Linux operative systems.

The interface of *MH-shiny* consists of three main sections (Fig. 6; Figs. S2–S4). The first section is a user-friendly "Interactive Map" where the user can select the country, region, site or site type directly on the map (Fig. S2). The selection will automatically connect to the second section, the "Data Explorer" which shows the data as a list of the literature items and a graphical representation of the variables (chosen by the user), which can be directly downloaded as a .csv file or as a .png image, respectively (Fig. S3). The third section "Keywords Analysis" provides a word-cloud analysis of keywords (author or index keywords) as a visual representation of the most frequently used, a graph of the number of occurrences of the keywords and a network graph of co-occurrence keywords (Fig. S4).

## DISCUSSION

### Existing knowledge on volcanic-associated ecosystems in the Mediterranean Sea

The whole database includes literature items published in more than 100 different journals. Among the top five journals by number of published studies on these subjects, only *Marine Environmental Research* included the analysis of biological aspects in its aims.
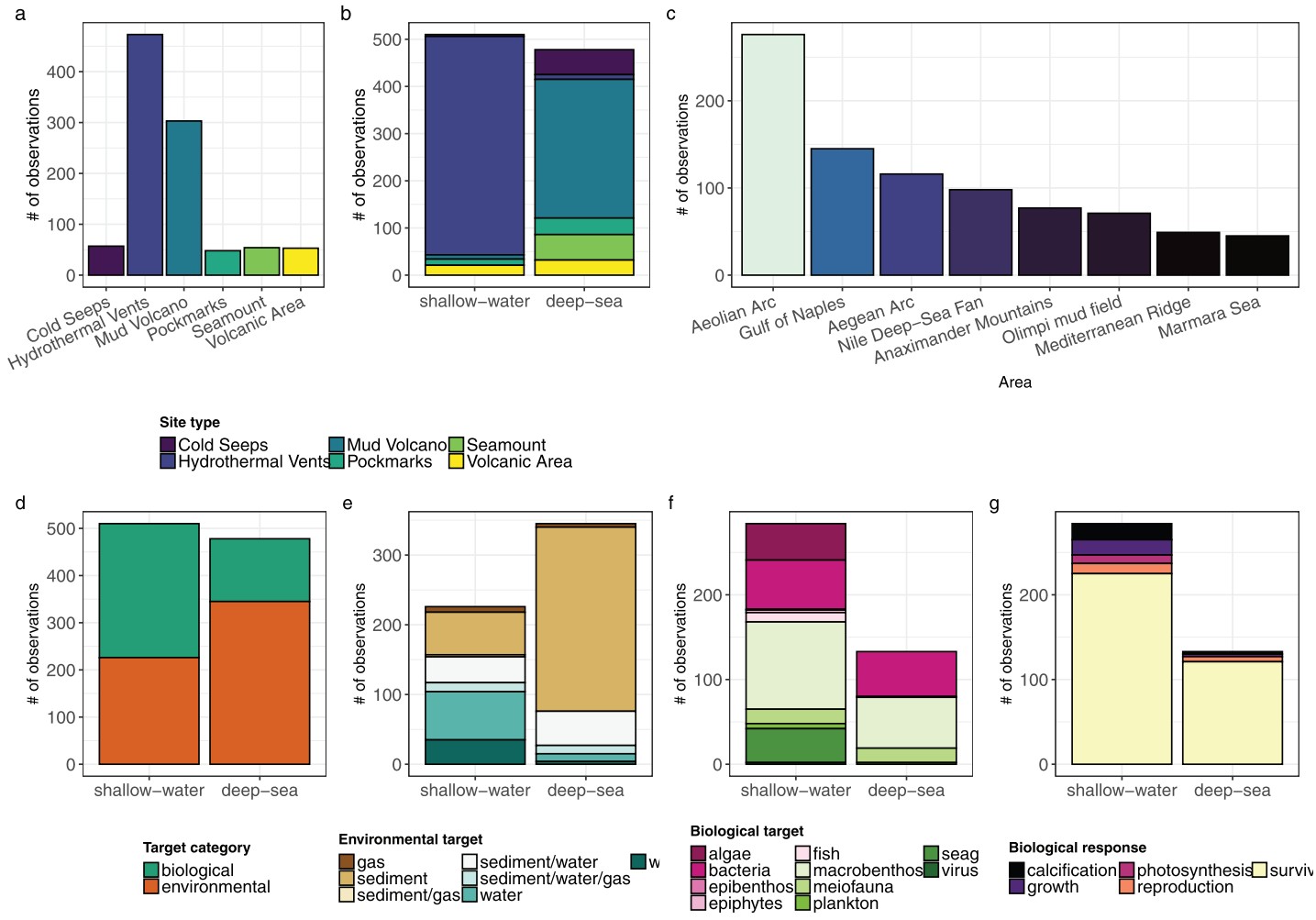

**Figure 4** **(Bio)geographic, population and biological responses variables distribution of all observations (n = 992).** (A) Number of observations per site type. (B) Number of observations per site type subdivided into shallow water and deep sea. (C) Number of observations in the different regions (only *n.* of observation > 30). (D) Number of observations per target category subdivided in shallow water and deep sea. (E) Number of observations per environmental target subdivided in shallow water and deep sea. (F) Number of observations per biological target subdivided in shallow water and deep sea. (G) Number of observations per biological responses subdivided in shallow water and deep sea.

Our database covers approximately 50 years of research dating back to 1973, with the first study focused on hydrothermal metalliferous deposits of Santorini Island in Greece (*Rydell & Bonatti, 1973*).

Since then, the scientific literature on volcanic-associated ecosystems has been characterised by an annual growth rate of 10%, with a peak in the year 2000. The number of published studies remained relatively constant until 2009, with most of the studies focusing on abiotic variables and only a few investigations including the analysis of biological components. From 2009, the number of published studies almost tripled in response to an increasing scientific interest towards shallow hydrothermal vents, after their use as potential natural laboratories for studying the effects of ocean acidification on marine ecosystems (*Hall-Spencer et al., 2008*). The study of *Hall-Spencer et al. (2008)* was a

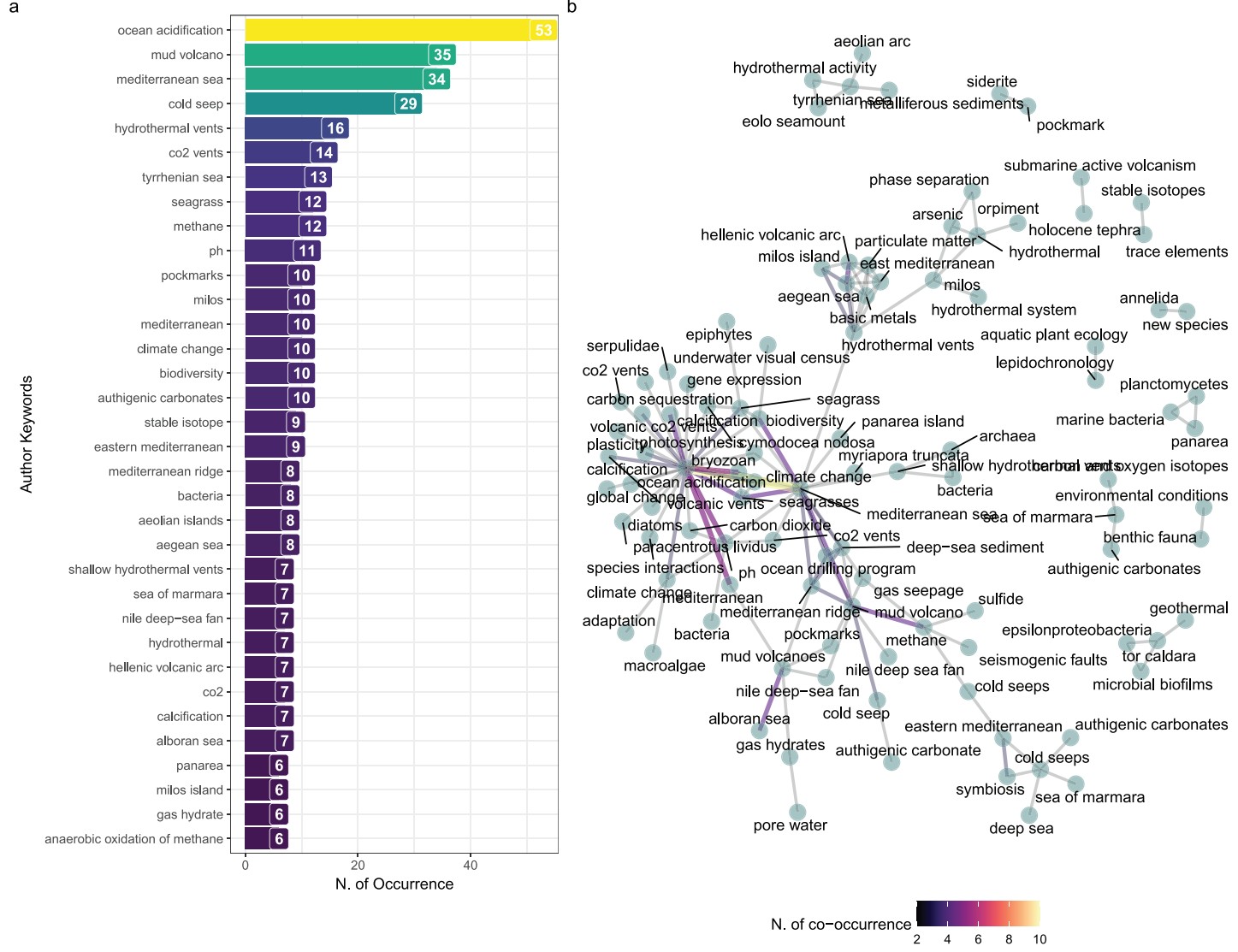

**Figure 5 Authors keywords occurrences and co-occurrence network.** (A) Number of occurrences of author keywords in the selected studies (*n* > 4). (B) Network visualization of co-occurrences. The thickness of the line indicates the number of co-occurrences between two single keywords.

turning point since it was the first to describe shallow hydrothermal vents as analogues of future acidified oceans, where $CO_2$ emissions naturally decrease the local pH exposing the nearby living organisms to environmental conditions likely similar to those expected in the future. Since then, studies on hydrothermal vents provided valuable insights on the potential mechanisms for adaptation and resilience in the face of changing ocean conditions (*Aiuppa et al., 2021*). Understanding these impacts and adaptations is crucial to undertake conservation and management actions to protect and maintain healthy ecosystems. The role of shallow hydrothermal vents for understanding the impacts of ocean acidification may explain why most studies in our database have focused on hydrothermal vents in shallow areas rather than other volcanic-associated ecosystems, despite pockmarks, seamount, mud volcanoes and cold seeps support a variety of

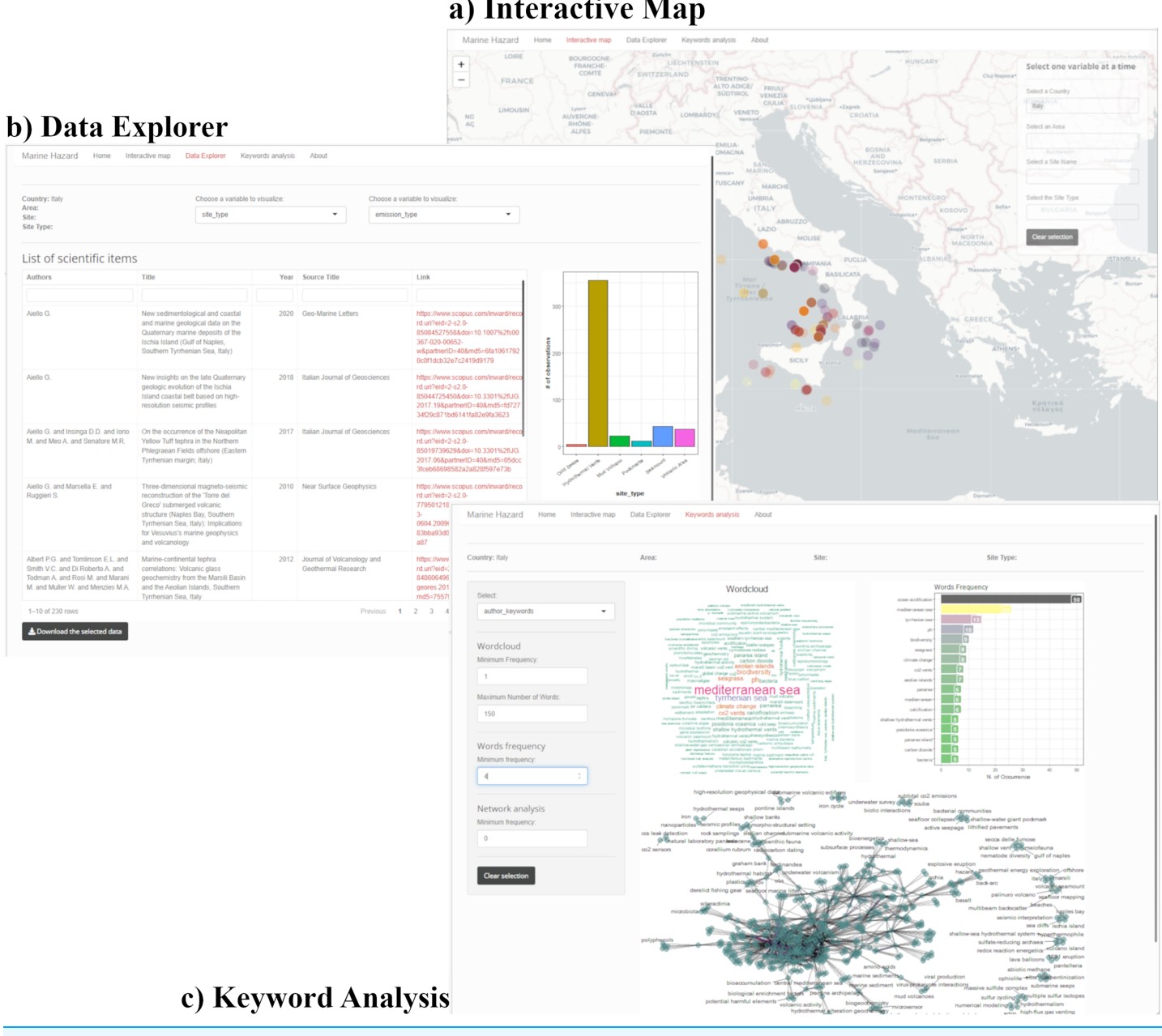

**Figure 6 Screenshots of the three main sections of the MH-shiny interface.** (A) "Interactive Map". (B) "Data Explorer". (C) "Keywords Analysis". Example with the selection of the country "Italy".

organisms including bacteria and other microorganisms as well as larger animals (*Olu-Le Roy et al., 2004*; *Taviani et al., 2013*; *Ingrassia et al., 2015*).

In our database, we identified 156 different unique sites across all the different volcanic-associated ecosystems and 21 different regions ranging in depth from less than 1 m (*e.g.*, Vulcano Island, Aeolian Archipelago, Italy) to 3,800 m depth (*e.g.*, the Cobblestone Area in the Mediterranean Ridge). Most of the scientific information available comes from research undertaken in the Aeolian Arc and in the Gulf of Naples (Italy), followed by the Aegean Arc (Greece). Regarding the Aeolian Arc, more than 100

observations were focused on just three sites: Panarea island and nearby islets, Vulcano island and the submerged volcano Marsili, with more than 75% of the observations conducted on hydrothermal vents.

The first study mentioning the hydrothermal vents in Panarea island was published in 1985 (*Beccaluva et al., 1985*). The number of studies in this island has increased rapidly after a strong degassing activity in the vent area that was firstly recorded in 2002 (*Capaccioni et al., 2005*) and is still ongoing. The area has been studied in terms of the effects of acidification on ecologically relevant organisms such as seagrasses, macrobenthic species and microorganisms (*e.g., Maugeri, Gugliandolo & Lentini, 2013*; *Esposito et al., 2015*; *Seebauer & Richert, 2017*).

Vulcano island constituted one of the study areas of the MedSea Project, a European project which ran from 2011 to 2014 and produced over 150 published studies on the effects of Ocean Acidification on marine organisms in the Mediterranean Sea (*Ziveri, 2015*). Many of them focused on biological targets such as seagrass, macroalgae and macrobenthos analyzing survival and calcification responses in the shallow hydrothermal vents on the Levante Bay, Vulcano Island (*e.g., Johnson et al., 2012*; *Boatta et al., 2013*; *Milazzo et al., 2014*; *Hendriks et al., 2014*).

In the Gulf of Naples, more than 50 studies were conducted in the shallow hydrothermal vents on Ischia Island (Italy). Ischia Island was the first site identified as a "natural laboratory" for studying the effects of low pH conditions on marine organisms, including the response of calcification rates in molluscs and corals (to name few among the most outstanding studies: *Hall-Spencer et al., 2008*; *Cigliano et al., 2010*; *Kroeker et al., 2011*; *Ricevuto et al., 2012*; *Kroeker, Gambi & Micheli, 2013*; *Gambi et al., 2016*).

According to our database, the biological categories that have been most thoroughly studied are macrobenthos, algae and bacteria. These studies have primarily focused on shallow water systems, rather than deep sea systems. The highest number of biological observations have been centered on the biological response to acidification, with a significant portion of the literature focusing on the issue of ocean acidification in the Mediterranean Sea, as indicated by the high number of occurrences of the keywords "ocean acidification" coupled with "mediterranean sea" in the keywords analysis.

Our investigation also highlighted substantial differences in the scientific research efforts towards volcanic-associated ecosystems in the Mediterranean Sea. Most studies focused on investigating the environmental set-up (morphological, geophysical and chemical characteristics) rather than the biological communities associated with these environments, especially in the deep sea compared to the shallow water. This is possibly related to the potential industrial exploitation such as oil and gas exploration in deep-sea areas, where pockmarks and mud volcanoes are sentinels for potential rocks source or reservoirs (*Loncke, Mascle & Fanil Scientific Parties, 2004*). In addition, the use of shallow hydrothermal vents for assessing the impact of ocean acidification has produced a plethora of studies in such ecosystems (see *Aiuppa et al., 2021* and reference therein).

## Protection of volcanic-associated ecosystems in the Mediterranean Sea

The high number of research studies in the Mediterranean Sea, that see volcanic-associated ecosystems as principal subjects, highlights the growing scientific value of these peculiar ecosystems, with hydrothermal vents resulting more explored than the rest of the site type. Nonetheless, the SM allowed us to highlight the low occurrence of protection and conservation measures including volcanic-associated ecosystems in the Mediterranean Sea. Despite almost 30% of the observations falls within 28 Area-Based Management Tools (ABMTs) for conservation (*i.e.*, instruments that manage areas by imposing stringent regulations delivered by a management authority to achieve high-level protection goals, *Gissi et al., 2022*), such as Marine Protected Areas, Nature Reserves or Parks, Special Areas for Conservation, Zone of Special Protection, Site of Community Importance, most of them are not protected by the ABMTs. All the ABMTs that enclose volcanic-associated ecosystems in Mediterranean Sea, have been established by European countries and are mainly located in shallow water in the central-western area of the Mediterranean Sea between Italy, Malta, France, Spain and Monaco, while the rest are located in Greece. All these ABMTs are subject to the European legislation transposed by each individual country. Fourteen sites are included in the Natura 2000, the largest coordinated network of protected areas in the world representing the strongest European legislative tool for the conservation of Europe's most valuable and threatened species and habitats, listed under both the Birds Directive (2009/147/EC) and the Habitats Directive (92/43/EEC). Hydrothermal vents and seamounts are listed in the Annex I of the Habitats Directive (92/43/EEC) within the habitat category 1,170 "Reefs" (PAL.CLASS.: 11.24, 11.25). In addition, hydrothermal vents along with pockmarks are listed in the same document as "bubbling reefs" and "pockmarks" respectively within the habitat category 1,180 "Submarine structures made by leaking gases" (PAL.CLASS.: 11.24) of the same Directive. Despite the presence of these habitats in the Directive, the sites in the Mediterranean Sea that are part of the Natura 2000 framework are not directly protected for this reason. Beyond these 14, only other 14 AMBTs mapped in this study enclosed volcanic-associated ecosystems and are subjected to local conservation measures. Among the sites identified in the eastern Mediterranean Sea, only one site named OCEANID off the west coast of Cyprus has been proposed as a Natura 2000 site (CY4000024 pSCI = proposed Sites for Community Importance).

The lack of protection measures for the majority of the volcanic-associated ecosystems mapped in our SM is likely due to the fact that many of them are not easily accessible due to their location (high depth and/or distance from the continent), limiting both their exploration and the feasibility of enforcement, monitoring and surveillance of any protection measure put in place (*Mazaris et al., 2018*). The user-friendly searchable database *MH-shiny* here developed can aid in the implementation of more effective protection measures for these volcanic-associated ecosystems in the Mediterranean Sea within existing management instruments under the EU Habitats Directive. Moreover, to

enhance the comprehensiveness of the SM, future updates could include multiple languages investigators to access a broader range of literature.

### The Aeolian Archipelago case study

Despite 30% of the volcanic-associated ecosystems are enclosed within ABMTs for conservation, not all of them are protected. To explain this discrepancy, we use the case study of the Aeolian Archipelago, a volcanic arc located in the southern Tyrrhenian Sea. This region is characterized by an exceptional marine biodiversity due to the elevated number of different habitat types and organisms present in the area (*Consoli et al., 2021*). Marine hydrothermalism is a diffuse phenomenon in the whole area and the entire volcanic arc is the most scientifically explored in the Mediterranean Sea because of the heterogeneity of its hydrothermal structures (*Dekov & Savelli, 2004*; *Lupton et al., 2011*; *Esposito et al., 2018*; *Rizzo et al., 2022*). The archipelago has an extension of 22 km$^2$ and consists of seven main islands (Alicudi, Filicudi, Salina, Lipari, Vulcano, Panarea, Stromboli) with associated islets (Basiluzzo, Dattilo, Bottaro, Lisca Bianca, Strombolicchio), and several seamounts (*Beccaluva et al., 1985*; *Gamberi & Marani, 1997*; *Lupton et al., 2011*). The area holds the status of Special Conservation Area (ITA030044 ZPS-RETE NATURA 2000 "Arcipelago delle Eolie—area marina e Terrestre") based on the EU Birds Directive (2009/147/EC) and is therefore identified as Natura 2000 site and has been listed as UNESCO World Heritage Site (*Angelini, 2008*). The establishment of a national Marine Protected Area that covers the entire archipelago, already planned by the Italian law 979/82, has been underway for decades. In 2014, the Sicily region in collaboration with UNESCO delivered a proposal of a management plan to overcome the mismanagement of the natural, geophysical and archaeological heritage of the archipelago, supporting the establishment of the Marine Protected Area (MPA) (*Angelini, 2008*).

We identified fourteen different important habitat types characterizing the volcanic-associated ecosystems in the area, along with several protected species of algae and invertebrates living in association with the hydrothermal vents (*UNEP, 1973*; *Council of Europe, 1979*) (Table S6). Among these habitats, several are listed as *priority habitats* in the Annex I of the Habitats Directive (92/43/EEC) and in the Protocol Concerning Specially Protected Areas and Biological Diversity in the Mediterranean of the Barcelona Convention (SPA/BD Protocol, *UNEP/MAP, 1995*; *Consoli et al., 2021*) (Table S6). According to the SPA/BD Protocol, the conservation of these habitats is mandatory (SPA/BD protocol of the Barcelona Convention, *UNEP/MAP, 1995*).

Despite the efforts to improve the protection of the marine environment surrounding the Aeolian Archipelago, the entire area is still subjected to significant anthropogenic pressures deriving especially from touristic and artisanal fishing activities on which the archipelago's economy strongly relies. Derelict fishing gears and general waste from land pollution and touristic activities (*e.g.*, plastic bottles, metals, ceramics, glass) have been found associated with hydrothermal sites around the islands where entanglement and ghost fishing have been documented also, providing the unquestionable evidence of the

anthropogenic impacts even in the deep sea (*Consoli et al., 2021*). Moreover, years of unmanaged scientific research that left instruments and/or used destructive sampling methodologies could represent a source of additional environmental disturbance in the area (*Dando & Juniper, 2001*).

Several investigation have been conducted in the archipelago providing data on the geochemical (*Italiano & Nuccio, 1991*; *Sedwick & Stuben, 1996*; *Capaccioni, Tassi & Vaselli, 2001*; *Price & Pichler, 2005*; *Italiano, 2009*; *Boatta et al., 2013*; *Price et al., 2015*) and biological (*Calosi et al., 2013*; *Apostolaki et al., 2014*; *Johnson et al., 2015*; *Harvey et al., 2016*; *Vizzini et al., 2017*; *Mirasole et al., 2020*; *Noè et al., 2020*) setup of many hydrothermal vent sites of this area. Here, marine hydrothermalism is characterized by sporadic and unpredictable underwater phenomena such as the 2002 massive underwater explosion near Panarea Island (*Esposito, Giordano & Anzidei, 2006*). Such natural hazard phenomena can severely impact the marine environment through the release of great amounts of heavy metals and trace elements in the surrounding habitats, potentially causing bioaccumulation in the local fishing population (*Andaloro et al., 2012*). This is a public safety concern that needs to be managed appropriately in order to avoid damage to the local population, fishers, divers, or tourists (*Esposito, Giordano & Anzidei, 2006*; *Aliani et al., 2010*; *Vizzini et al., 2013*; *Mishra, Santos & Hall-Spencer, 2020*; *Voltattorni et al., 2006*).

Considering the detrimental impacts that the above-described commercial and scientific activities can have on morphological, geochemical, and biological aspects of the ecosystems along with the occurrence of protected species and priority habitats, the gas hazard and the related environmental contamination risk, the setting and implementation of management measures in line with the current conservation policies in this region is urgently needed to protect and manage these peculiar ecosystems.

The creation of a MPA (according to Italian Law 979/82, Art. 31) where all activities are strictly regulated could be a solution, with no-take/no-access zones regimes (*i.e.*, integral reserve) at least for the most sensitive hydrothermal sites (*Esposito et al., 2018*; *Aiuppa et al., 2021*; *Consoli et al., 2021*). The designation of Site of Community Importance (*SIC*) or Special Area of Conservation (SAC) would also be legitimated by the presence of critical habitat types or species as respectively listed in Annex I and II of the European Habitats Directive (http://ec.europa.eu/environment/nature/legislation/habitatsdirective/index_en.htm#enlargement). The MPA would be the best bet to manage multiple uses of the archipelago with conservation as a priority objective, followed by the allocation of areas for regulated scientific research and monitoring, in light of the strong need to fill knowledge gaps on the present volcanic ecosystems. Implementing up-to-date protection measures in the Aeolian Archipelago would also increase people's awareness of the importance of preserving marine hydrothermal vents based on their high naturalistic importance, thereby encouraging the development of sustainable activities for fishing and tourism in the whole area. An ecosystem-based conservation strategy is required to identify the priority criteria for the protection of these volcanic-associated ecosystems (*Fanelli et al., 2021*).

### Limitations of our systematic map

It is important to highlight some methodological limitations in our protocol and some limitations based on the literature investigated, that could be however addressed in subsequent updates.

For instance, due to finite time and resources, we were unable to conduct additional searches on other databases and our analysis of grey literature was limited to the first 100 items found on Google Scholar. We were also restricted to accessing articles, documents, and reports available online and limited to English language, and our analysis was limited to the Mediterranean region.

However, despite these limitations, our Systematic Map provides an indication of the robustness of the evidence, based on the protocol adopted. While it does not provide a detailed quality appraisal of the articles or how they address susceptibility to biases and heterogeneity of effects, it is a valuable resource for understanding the current state of knowledge about volcanic-associated ecosystems in the Mediterranean Sea. It is hoped that future updates will be build on this initial work and provide a more comprehensive overview of this important area of research.

## CONCLUSIONS

Beyond the limitation of our systematic map as discussed above, this study aimed to summarize the current state of knowledge and protection of volcanic-associated ecosystems in the Mediterranean Sea, including hydrothermal vents, cold seeps, pockmarks and seamounts, to address future research efforts and inform conservation and protection initiatives. Our Systematic Map (1) summarized the existing knowledge on volcanic-associated ecosystems, including hydrothermal vents, cold seeps, pockmarks and seamounts in the Mediterranean basin, and (2) created a user-friendly, free and searchable database in the form of a *Shiny* web-based application. The database enclosed 433 literature items covering approximately 50 years of scientific research. It highlighted the higher number of studies involving environmental characteristics of the volcanic-associated ecosystems, probably driven by exploitation and economic interests, while the biological studies started only in the last two decades with a main focus on understanding the effects of ocean acidification. The results of this investigation show that despite the high scientific importance and ecological and economic value of volcanic-associated ecosystems, as well as their consideration in international conservation policies applied in the Mediterranean Sea (*e.g.*, Habitats Directive), they are still inadequately protected. More specific protection measures, implemented in both new and existing Area-Based Management and conservation tools are needed.

Our *MH-shiny* web-based application and interactive map offer a tool for policymakers to narrow the gap between research evidence and environmental management in the context of the FAIR principles of the European Commission. Our code is freely available and may be easily updated and re-analyzed. The searchable database in our *MH-shiny* can help the implementation of ecosystem-based management plans informing decision-makers, stakeholders and public opinion in taking evidence-based decisions.

### Funding

This work was funded by PON "R&C" 2007-2013-PON03PE_0023_1 Project "Marine Hazard", Italian Ministry of University and Research. The funders had no role in study design, data collection and analysis, decision to publish, or preparation of the manuscript.

### Grant Disclosures

The following grant information was disclosed by the authors:
PON "R&C" 2007-2013-PON03PE_0023_1 Project "Marine Hazard".
Italian Ministry of University and Research.

### Competing Interests

Valentina Costa is a PeerJ Hub Editor for the IABO Hub.

### Author Contributions

- Valentina Costa conceived and designed the experiments, performed the experiments, analyzed the data, prepared figures and/or tables, authored or reviewed drafts of the article, and approved the final draft.
- Valentina Sciutteri performed the experiments, analyzed the data, prepared figures and/or tables, authored or reviewed drafts of the article, and approved the final draft.
- Pierpaolo Consoli analyzed the data, authored or reviewed drafts of the article, and approved the final draft.
- Elisabetta Manea analyzed the data, authored or reviewed drafts of the article, and approved the final draft.
- Elisabetta Menini analyzed the data, authored or reviewed drafts of the article, and approved the final draft.
- Franco Andaloro analyzed the data, authored or reviewed drafts of the article, and approved the final draft.
- Teresa Romeo analyzed the data, authored or reviewed drafts of the article, and approved the final draft.
- Roberto Danovaro analyzed the data, authored or reviewed drafts of the article, and approved the final draft.

### Data Availability

The data and the complete R code are available at GitHub and Zenodo: https://github.com/costavale/MH-shiny; Valentina Costa. (2023). costavale/MH-shiny: MH-shiny app (v1.1). Zenodo. https://doi.org/10.5281/zenodo.7663999.

### Supplemental Information

Supplemental information for this article can be found online at http://dx.doi.org/10.7717/peerj.15162#supplemental-information.

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
