# Peer review of "Volcanic-associated ecosystems of the Mediterranean Sea: a systematic map and an interactive tool to support their conservation"

_PeerJ, doi:10.7717/peerj.15162_

## Round 0.1 · original submission · Minor Revisions

The three reviewers I have consulted all expressed a positive opinion about the paper and, overall, they offer suggestions for fairly minor revisions, which I kindly ask you to take into account before you resubmit the paper for further consideration.

In addition to the reviewers' comments, I would also like to encourage you to include in the paper some additional reflections about potential biases in your study, both at the methodological level (e.g., biases that may arise from excluding grey literature from the search) as well as in the investigated literature itself (e.g., you observe more studies in shallow waters than in deep sea; does this reflect a real-world distribution of the investigated ecosystems or is it merely a result of the higher ease with which ecosystems in shallow waters can be reached and studied by the researchers?).

·

Basic reporting

This is a rigorous and useful contribution to the scientific literature on hydrothermalism and marine conservation in the Mediterranean

Experimental design

No comment

Validity of the findings

This is a very thorough study and the findings are valid

Additional comments

Well stated conclusions, it is true that the hydrothermal habitats mentioned are not properly cared for and managed in the Mediterranean context.

Reviewer 2 ·

Basic reporting

The manuscript by Costa et al. proposes a systematic map developed to (1) identify the available scientific literature related to volcanic-associated ecosystems in the Mediterranean, (2) categorise and compare the available scientific information from different Mediterranean regions in terms of variables extracted from the literature (e.g., geographic information and populations analysed in the literature), (3) create a user-friendly and interactive map linked to a searchable database to support the translation of science into policy and management. The main goal is to provide a tool for future research efforts and conservation initiatives in the Mediterranean. What I find interesting is the collection of all the literature on these ecosystems in the Mediterranean and the creation of the map that is, to my knowledge, the first to date for the Mediterranean, and will be certainly useful for comprehensive protection measures and management plans. Indeed, as the authors note, the unique biological processes and communities, and the important role they play in understanding the future of our oceans, are a reason why volcanic-associated ecosystems must be protected also in the Mediterranean Sea. I believe the study in the complex is well suited for publication in your journal and I recommend minor revisions.
The topic and objectives of the paper, as well as the target audience, are clearly outlined. However, considering that almost all international directives mentioned by the authors focus on the deep volcanic-associated ecosystems, usually located along the oceanic ridge (> 1000 m depth), I suggest adding more information about the importance of the shallow volcanic ecosystems of the Mediterranean and why they must be considered as an environment worth protecting.

Experimental design

The methods used to achieve the objectives of the work are coherent and clearly explained and are well supported by the supplementary material. However, in my opinion to make the app more attractive and understandable, and to help policy makers in habitat conservation, it would be good if more information was added to the interface, if possible, for example on the presence of protected or rare species and habitats.
The subsections are coherent and well organized to help the reader follow all the steps of a methodology that is not really easy to understand.

Validity of the findings

All sections of the review are well developed. The methodology chosen for the literature review, the analysis of key words, the results obtained, and the discussion and conclusions all seem to be well aligned with the goals set forth in the introduction.
The conclusions of the study are supported by the results obtained, which show some gaps in biological research, mainly focused on the effects of ocean acidification, and in conservation policies applied to these ecosystems in the Mediterranean Sea, as well as the need for the developed MH-shiny web-based application and interactive map to bridge the gap between research results and environmental management.

Additional comments

Please take into consideration my correction on your manuscript (in the attached file)

Annotated reviews are not available for download in order to protect the identity of reviewers who chose to remain anonymous.

Reviewer 3 ·

Basic reporting

The manuscript by Costa et al. entitled "Volcanic-associated ecosystems of the Mediterranean Sea: a Systematic Map and an Interactive Tool to support their conservation" presents a systematic map and an interactive tool to aid future conservation actions of volcanic-associated marine ecosystems in the Mediterranean Sea. These ecosystems, which include hydrothermal vents, cold seeps, pockmarks, and seamounts, are widely distributed on the ocean floor and have been the subject of increasing scientific interest in the past decades. However, the available information on these ecosystems is limited, scattered, and unsuitable for supporting decision-making processes for their conservation and management.

Experimental design

The methods are well described with good details. The authors searched the Scopus database and the platform Web of Science to collect scientific information available for these ecosystems in the Mediterranean Sea. The collected literature and the bio-geographic and population variables extracted are provided into a Systematic Map as an online tool that includes an updated database searchable through a user-friendly R-shiny app. As far as I can see and if my attempt to use the app worked well, one weakness could that the app allows to display a single variable at a time. Is it possible to adjust that, giving users the opportunity to overlap variables?

Validity of the findings

The results provide valuable insights into the topic with important implications for future research and conservation efforts for these ecosystems in the Mediterranean Sea.
However I ask authors to improve the geochemical contents or at least to provide evidence in the discussion of the many studies available for many volcanic sites

Additional comments

My suggestion is minor revisions asking the authors the possibility to fix the variables' display and importantly to update the literature in the focus paragraph on the Aeolian archipelago, where many biological and ecological studies on these ecosystems were recently published.
In this paragraphs, I noticed a high rate of self-citations (i.e., many papers from co-authors of this ms are cited) and a slight bias towards the biological studies.
In this regard, I would suggest authors at least to dig the review paper by Aiuppa et al (already cited in their manuscript) in which geochemical data are examined in detail and a meta-analysis on biological is also provided with many more recent papers on this topic.

---

## Round 0.2 · accepted · Accept

The authors have satisfactorily addressed all of the reviewers' comments and the manuscript is ready for publication in PeerJ.